

# Three-dimensional variations in the lower limb caused by the windlass mechanism

María José Manfredi-Márquez[1],[*], Natalia Tovaruela-Carrión[1], Priscila Távara-Vidalón[1], Gabriel Domínguez-Maldonado[1], Lourdes María Fernández-Seguín[2] and Javier Ramos-Ortega[1],[*]

[1] Department of Podiatry, Universidad de Sevilla, Seville, Spain
[2] Department of Physiotherapy, Universidad de Sevilla, Seville, Spain
[*] These authors contributed equally to this work.

## ABSTRACT

**Background:** The windlass mechanism was described as the effect caused by the extension of the first metatarsalphalangeal joint (1st MTPJ). Quantify the degrees of movement produced in the leg by means of the Bioval® sensor system, after performing two measurements in the 1st MTPJ, 45° extension and maximum extension.

**Methods:** Tests-post-test study with just one intervention group, performed in the Clinical Podiatry Area of the Faculty of Nursing, Physiotherapy and Podiatry of the University of Seville. Subjects were included as of age 20, with a value from 0° to 3° valgus, Helbing line, a value from 0° to +5° for the foot postural index, and a localisation axis for the normalised subtalar joint. Subjects with surgical operations of the first ray, fractures and surgical operations in the leg, pathologies in the first ray and rheumatic diseases were excluded. Measurement was performed with the Bioval® system by means of inserting four sensors in the bone structures involved in the windlass mechanism.

**Results:** With the 45° wedge we observed a direct correlation among the variables extension–plantar flexion 1st MTPJ and rotation of the femur. With maximal extension of the 1st MTPJ we obtained a direct relationship between the variable extension of the 1st MTPJ and the variables plantar flexion and prono-supination of the 1st metatarsal as well as with the variables tibia rotation and femur rotation.

**Conclusion:** Kinematic analysis suggested that the higher the degree of extension the more movement will be generated. This reduces the level of impact the more distal the structure with respect to the 1st MTPJ, which has an impact on the entire leg. Because of the kinematic system used wasn't suitable, its impact wasn't exactly quantified.

Corresponding authors
María José Manfredi-Márquez, marmanmar1@alum.us.es
Javier Ramos-Ortega, jrortega@us.es

## INTRODUCTION

In 1954 *Hicks (1954)* reported the mechanism by which metatarsalphalangeal extension raise the medial longitudinal arch (MLA) by tensing the plantar fascia (PF) during propulsion without the assistance of muscular action. This effect is called 'windlass

mechanism' (WM). In his study, it was observed that during passive extension of the hallux, the PF wraps rounds the head of the first metatarsal, increasing its tension and increasing the height of the MLA, thereby reducing the distance between its origin and insertion. Moreover, the increased arch will cause supination of the hindfoot and external rotation of the lower limb. Finally, Hicks revealed the important relationship set out between the PF and first metatarsalphalangeal joint (1st MTPJ). These are the most important players in the implementation of this mechanism, where the subtalar joint (STJ), midtarsal joint (MTJ) and the ankle joint (AJ), in addition to the tibia and femur (*Hicks, 1954*; *Durrant, 2009*; *Harradine, Bevan & Carter, 2006*; *Harton, Weiskopf & Goecker, 2002*; *Kirby, 1997*; *López, 2012*; *Paton, 2006*) are also involved.

The first metatarso-digital segment is essential for correct function of the WM, because this is activated with passive extension of the 1st MTPJ (*Hicks, 1954*).

Extension of the 1st MTPJ and subsequent plantar flexion of the first ray will lead to MLA elevation by means of increased tension of the PF. Its length reduces and there is modification of the position of all the joints that take part in this mechanism, generating a supination movement in the STJ capable of producing a change in position in the cyma line, where the talonavicular joint move from a location anterior to the calcaneocuboid joint to a posterior position (*Fuller, 2000*; *Munuera, 2009*).

At the same time a dorsiflexion movement of the talus occurs inside the AJ which helps to stabilise the foot during gait and external rotation of the lower extremity (*Munuera, 2009*). Finally, the last movement produced is supination of the MTJ around its oblique axis that avoids elevation and abduction of the forefoot, keeping it anchored to the ground (*Fuller, 2000*; *Munuera, 2009*). The movements produced in the WM are triplanar, not only in the sagittal plane (*Fuller, 2000*).

In short, when the arch is raised, the first metatarsal plantar flexes, STJ and the oblique axis of the MTJ supinate, tibia and femur externally rotate and the pelvis moves backwards. Therefore, the foot movement is translated into an ascending chain in the tibia, femur and pelvis (*Hicks, 1954*; *López, 2012*) giving the foot as a whole the necessary stability during the ultimate support phase (*Hicks, 1954*; *Kirby, 1997*; *Fuller, 2000*; *Munuera, 2009*; *Aquino & Payne, 2001*; *Bolgla & Malone, 2004*; *Song et al., 2013*).

The WM produces a rotation of the lower limb, but it isn't known to what extent this relationship occurs. It is necessary to know how many degrees of tibial rotation occur when the 1st MTPJ is extended, so that this rotation can be applied in the control of pronation during mould making of the phenolic foam in weight-bearing.

The aims of this study are to quantify the degrees of movement produced in the lower limb by means of the Bioval® sensor system, after performing two measurements in the 1st MTPJ, 45° extension and maximum extension.

## MATERIALS AND METHODS

### Subjects

Tests-post-test study with just one intervention group, performed in the Clinical Podiatry Area of the Faculty of Nursing, Physiotherapy and Podiatry at the University of Seville

(Spain) between November 2015 and April 2016. Subjects were selected using convenience sampling; in this case it was subjects belonging to the Faculty of Nursing, Physiotherapy and Podiatry at the University of Seville, considering as inclusion criteria aged over 20. This age threshold was selected because of understanding that up to 17–19 years, in some cases up to 20 years, bone growth physis has not yet closed, which enables more capacity for torsional changes in these segments (*Shapiro & Forriol, 2005*); a value of 0–3° valgus in the Helbing line (*Sell et al., 1994*), a value of 0 to +5 for the foot posture index (FPI) (*Pascual et al., 2013*; *Redmond, Crosbie & Ouvrier, 2006*) and a normalised STJ localisation axis (*Kirby, 2001*). Exclusion criteria were subjects with surgical operations of the first ray, fractures and surgical operations in the leg, pathologies in the first ray (Hallux Limitus, Hallux Rigidus, Hallux Valgus), rheumatic diseases, refusal to sign the informed consent form. All those who offered to take part in the study gave their informed consent in writing before being recruited. The ethical and legal principles required in any biomedical research essentially considered in the Oviedo Agreement and the Declaration of Helsinki were compliant at all times; in addition there was guarantee of data confidentiality complying with Spanish Law 4/2002, of 14 November, basic regulator of patient autonomy (*AMM, 1964*; *Consejo de Europa, 1997*).

## Methodology

A data collation form was designed; the first step was to record the sex and age of subjects. Some shorts then provided to facilitate performing measurements set out in our protocol.

Subjects subsequently removed their shoes and their weight and height were measured with Mechanical Column Scale with Tallimeter (Seca 711. Class III). The Helbing's line (*Sell et al., 1994*) and the FPI (*Pascual et al., 2013*; *Redmond, Crosbie & Ouvrier, 2006*) were measurement with the subject standing on a podoscope.

Extension of the 1st MTPJ was measured with the subject in supine position to verify the movement of this (*Munuera, 2009*). Subsequently, the STJ axis location was done using the palpation technique reported by *Kirby (2001)*. This technique was made by one research with 10 years of experience. After scanning the footprint with the Plantar Digital Scanner CbsScanFoot model EDP-G2-A with the location axis of STJ marked, the degrees of this axis were quantified (*Kirby, 2001*) using longitudinal bisection of the foot. For this, the software Autocad® (Autodesk Inc., San Rafael, CA, USA) (*Munuera, 2006*; *Munuera et al., 2006*; *Ramos et al., 2014*) was used.

Measurement was then performed with the Bioval System (RM Ingénierie, Rodez, France) (*Grand & Geronimi, 2011*). This is a system of biomechanical analysis (or analysis of human movement), that allows to value, visualise and quantify the movement in the three planes of the space; sagittal plane (flexion/extension), frontal (adduction/abduction) and transverse (internal/external rotation) of all joints in static and dynamic with a frequency of 30 Hz.

This system is based on the use of four inertial sensors fixed to the body, which transmit the information via Bluetooth, and represent it in the form of graphs.

The sensors may be configured in a dependent or independent manner. Dependent sensors value a joint by reference to the sensor that precedes it, requiring at least two

sensors (*Zurita, 2013*). The studied variables were: extension–flexion 1st MPTJ (yellow sensor), inversion/eversion 1st MTT (red sensor), tibia rotation (blue sensor) and femur rotation (green sensor). Independent sensors record isolated movement. The studied variables were: Extension 1st MPTJ (yellow sensor), plantar flexion 1st MTT and inversion/eversion 1st MTT (red sensor), tibia rotation (blue sensor) and femur rotation (green sensor).

The four sensors comprising it were previously placed: one on the dorsal area of the proximal phalanx of the first toe (yellow sensor), two on the medial area of the diaphysis of the first metatarsal (red sensor), three on the anterior tibial tuberosity (blue sensor) and four on the femur greater trochanter (green sensor) to observe and understand the movement along the three space planes.

Bioval takes a reference position from the relaxing standing position of the subject.

Once the sensors were in position, the Bioval® system was programmed in the computer and the measurement was taken in two ways:

1. Extension of the 1st MTPJ using a wedge with a 45° angle with dependent sensors: To perform this measurement, we had to previously configure the Bioval® system such that four sensors were interconnected, interdependent on one another; inter-movements were quantified. Measurement commenced when we raised the first toe, with the toe in relaxed position, continued whilst we placed this under an ethylene vinyl acetate wedge with a 45° angle and ended when we removed the wedge (Fig. 1). Among all the values obtained with this measurement, we took the maximum value as a reference.

2. Extension 1st MTPJ with independent sensors: In this case, the Bioval® system must present an independent system, quantifying in an isolated manner the degree of movement for each one of the sensors. In this case, the 1st MTPJ was extended to maximum twice, consecutively, with a time interval of 5 s between each one of them. Measurement commenced when we raised the first toe, with the toe in relaxed position and ended when we placed the first toe in a relaxed position (Fig. 2). Among all the values obtained with this measurement we took two values as reference. First, the value similar or equal to 45° extension and second, the maximum value.

All these measurements were performed on the right foot; sensor movement was recorded for 30 s.

The variables included in this study were classified as descriptive (sex, age, weight, height, BMI, Helbing line, FPI and STJ localisation axis) dependent (plantar flexion and inversion/eversion of the 1st metatarsal, rotation of the tibia and femur) and independent (extension of the 1st MTPJ).

## Statistical methods

A descriptive analysis of the variables included in the study was performed based in the distribution of the sample using the Shapiro–Wilks test. To determine whether statistical significance was attained we will use Student's *t*-test or the Wilcoxon

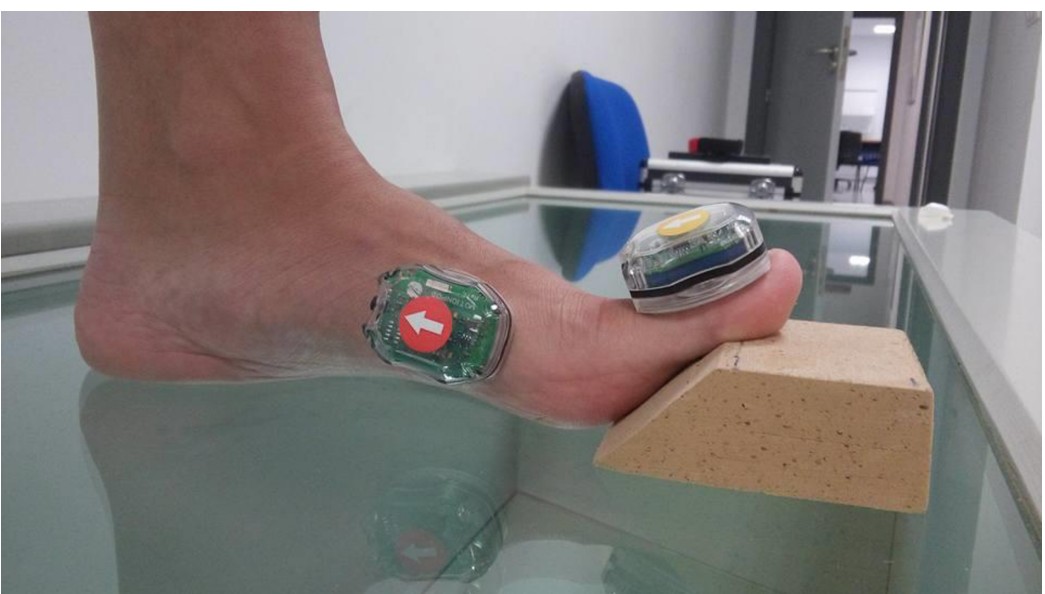

**Figure 1 Extension 1st MTPJ with the 45° angle wedge.** Photo credit: María José Manfredi–Márquez.

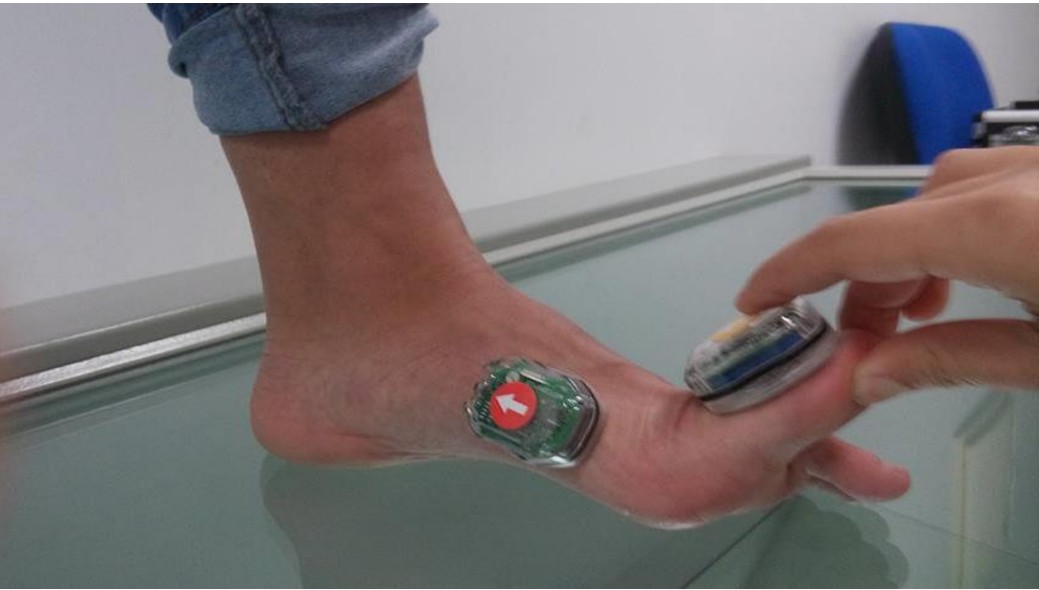

**Figure 2 Extension 1st MTPJ.** Photo credit: María José Manfredi–Márquez.

rank sum test for two related samples and to evaluate the correlation between variables Pearson test or Spearman's Rho test will use. Data analysis was performed using the programme Statistical Package for the Social Sciences version 22.0 (SPSS Inc., Chicago, IL, USA). A significance level of 0.05 was used for all statistical procedure.

## RESULTS

The study population was comprised of 15 subjects ($N = 15$), 11 women and four men aged between 20 and 27 years.

Results were analysed according to the set order in the data collection.

We analysed the distribution of dependent and independent variables for each study situation by means of the Shapiro–Wilks test (Table 1); a non-normal distribution predominated.

Because this is a non-normal distribution, the median and interquartile range were analysed for each variable, in addition to maximum and minimum value to measure the amount of data variation or dispersion (Tables 2–4).

To determine the existence of statistically significant differences between the recorded values with 45° extension and maximum values of the joint for each variable with independent sensors, we used the Wilcoxon rank sum test where we found statistically significant differences for extension of the 1st MTPJ, inversion/eversion and plantar flexion of the 1st metatarsal (Table 5).

To study the correlation existing among the different variables with 45° extension and maximum values of the joint, we used Spearman's Rho test. For the situation in which the sensors were related, the results revealed a direct correlation among the variables extension–plantar flexion of the 1st MTPJ and rotation of the femur, and inverse correlation between the latter and the variable inversion/eversion of the 1st metatarsal (Table 6).

In the situation which the sensors were independent and in 45° extension, no correlation whatsoever was found between the variables.

Finally, the results obtained in the sensors with independent configuration and maximal extension of the 1st MTPJ, revealed a direct relationship between the variables extension of the 1st MTPJ and plantar flexion of the 1st metatarsal, in addition to extension of the 1st MTPJ and inversion/eversion of the 1st metatarsal. Correlation was also obtained between the variables plantar flexion and inversion/eversion of the 1st metatarsal, inversion/eversion of the 1st metatarsal and rotation of the femur and among the variables tibia and femur rotation (Table 6).

## DISCUSSION

This study aims to quantitatively determine in subjects that comply with set inclusion and exclusion criteria, angular changes that occur in the leg during onset and development of the WM passively, with a determined extension of the 1st MTPJ using an inertial sensor system.

Moreover, we provide novel information for the current literature because we related and quantified, using kinematic analysis, movements produced after activation of the WM. These movements allude to the entire leg, not only the tibia (*Hicks, 1954*; *Fuller, 2000*; *Munuera, 2009*; *Bolgla & Malone, 2004*; *Chana, 2013*; *Kappel-Bargas et al., 1998*), but also the femur.

The results obtained reveal that after extending the 1st MTPJ by means of a 45° wedge with the sensors configured in a dependent manner, median extension of this joint was

**Table 1 Normality tests.**

**Shapiro–Wilk**

| | Related sensors 45° extension | Independent sensors and 45° extension | Independent sensors and maximum extension |
|---|---|---|---|
| Extension 1st MTPJ | <0.01 | <0.01 | 0.96 |
| Plantar flexion 1st MTT | – | 0.18 | 0.75 |
| Inversion–eversion 1st MTT | <0.01 | <0.01 | <0.01 |
| Tibia rotation | 0.04 | 0.01 | 0.09 |
| Femur rotation | <0.01 | <0.01 | <0.01 |
| Location STJ axis | 0.01 | 0.01 | 0.01 |

Notes:
Shapiro–Wilk—extension 1st MTPJ with EVA-wedge 45°—dependent sensors. Extension 1st MPTJ—independent sensors.
1st MPTJ, First metatarsalphalangeal joint; EVA, ethylene vinyl acetate; 1st MTT, First metatarsal; STJ, subtalar joint.

**Table 2 Descriptive statistic results.**

| | Extension—PFI 1st MPTJ | Inversion–eversion 1st MTT | Tibia rotation | Femur rotation | STJ |
|---|---|---|---|---|---|
| Median | 8.83 | 2.80 | 0.27 | 0.42 | 8 |
| Minimal | 5.24 | 0.18 | 0.01 | 0.03 | 7 |
| Maximum | 37.64 | 35.71 | 1.77 | 3.72 | 14 |
| Interquartile range | 14.13 | 9.80 | 0.87 | 0.71 | 3 |

Notes:
Extension 1st MPTJ with EVA-wedge 45°—dependent sensors.
1st MPTJ, First metatarsalphalangeal joint; EVA, ethylene vinyl acetate; 1st MTT, First metatarsal; STJ, subtalar joint; PFI, plantar flexion.

**Table 3 Descriptive statistics results.**

| | Extension 1st MPTJ | PFI 1st MTT | Inversion–eversion 1st MTT | Tibia rotation | Femur rotation | STJ |
|---|---|---|---|---|---|---|
| Median | 44.99 | 3.67 | 2.06 | 0.3 | 0.52 | 8 |
| Minimal | 34.31 | 1.03 | 0.01 | 0.06 | 0.02 | 7 |
| Maximum | 45.21 | 6.63 | 28.64 | 1.69 | 4.70 | 14 |
| Interquartile range | 3.37 | 1.95 | 2.96 | 0.77 | 1.85 | 3 |

Notes:
Extension 45° 1st MTPJ. Independent sensors.
1st MPTJ, First metatarsalphalangeal joint; 1st MTT, First metatarsal; STJ, subtalar joint; PFI, plantar flexion.

**Table 4 Descriptive statistic results.**

| | Extension 1st MPTJ | PFI 1st MTT | Inversion–eversion 1st MTT | Tibia rotation | Femur rotation | STJ |
|---|---|---|---|---|---|---|
| Median | 47.59 | 4.12 | 2.08 | 0.49 | 1.04 | 8 |
| Minimal | 34.31 | 1.03 | 0.51 | 0.02 | 0.17 | 7 |
| Maximum | 63.03 | 7.78 | 28.64 | 1.69 | 4.7 | 14 |
| Interquartile range | 11.55 | 3.06 | 3.07 | 0.67 | 0.87 | 3 |

Notes:
Maximum extension of the 1st MPTJ. Independent sensors.
1st MPTJ, First metatarsalphalangeal joint; 1st MTT, First metatarsal; STJ, subtalar joint; PFI, plantar flexion.

**Table 5 Wilcoxon rank sum test.**

|  | Extension_max—extension_IAMTF | PF1_max—PF1 | Inversion–eversion_max—inversion–eversion_1mtt | Rot_tib_max—Rot_tib | Rot_femur_max—Rot_femur |
|---|---|---|---|---|---|
| Z | −2.803 | −2.803 | −2.528 | −1.079 | −1.428 |
| Sig. asintót. (bilateral) | 0.005 | 0.005 | 0.011 | 0.281 | 0.153 |

Note:
Extension 45°—maximal for the 1st MPTJ. Independent sensors.

**Table 6 Correlation test: Spearman's Rho.**

|  | Extension 1st MTPJ with 45° wedge—dependent sensors | | | | Maximum extension 1st MTPJ—independent sensors | | | | |
|---|---|---|---|---|---|---|---|---|---|
|  | Extension—PFI MTPJ | Inversion–eversion 1st MTT | Tibia rotation | Femur rotation | Extension 1st MPTJ | PFI 1st MTT | Inversion–eversion 1st MTT | Tibia rotation | Femur rotation |
| **Extension 1st MTPJ** | 1 | 0.04 | −0.14 | 0.57 | 1 | 0.66 | 0.43 | 0.20 | 0.16 |
| **PFI 1st MTT** | – | – | – | – | 0.66 | 1 | 0.63 | 0.01 | 0.26 |
| **Inversion–eversion 1st MTT** | 0.04 | 1 | −0.24 | −0.35 | 0.43 | 0.63 | 1 | 0.19 | 0.39 |
| **Tibia rotation** | −0.14 | −0.24 | 1 | 0.07 | 0.20 | 0.01 | 0.19 | 1 | 0.45 |
| **Femur rotation** | 0.57 | −0.35 | 0.07 | 1 | 0.16 | 0.26 | 0.39 | 0.45 | 1 |

Notes:
Extension 1st MTPJ with EVA-wedge 45°—dependent sensors maximal extension 1st MTPJ.
1st MPTJ, First metatarsalphalangeal joint; EVA, ethylene vinyl acetate; 1st MTT, First metatarsal; STJ, subtalar joint; PFI, plantar flexion.

8.83° ± 14.13, a value very far removed from the cases of 45°. We believe that this result may have been undermined by not considering the soft tissues that make up this joint. Continuing this measurement, the median obtained for the remaining variables are not very significant and almost negligible, leading to 2.80° ± 9.80 in inversion–eversion, 0.27° ± 0.87 in the tibia and 0.42° ± 0.71 in the femur. As we observed, there were fewer changes the further we moved from the WM main joint.

If we compare our results in maximum extension with the literature consulted, we verify that our value is 47.59°. Authors such as *Nester et al. (2014)*, *Halstead & Redmond (2006)* and *Nawoczenski, Baumhauer & Umberger (1999)* consider this value within the normal range of motion of the 1st MPTJ. However, Root (*Munuera, 2009*) considers that at least 60° extension is necessary to attain correct plantar flexion of the 1st metatarsal (*Munuera, 2009*). In addition to 10° plantar flexion, according to Root (*Munuera, 2009*; *Root, Orien & Weed, 1977*), and 22°, according to *Fuller (2000)* to enable full extension of the toe during the propulsive phase of gait, a value not attained in our measurement, which was 4.12°.

Results were unexpected. Probably the data recorded by the system reflect the actual situation. The measurement we took is indirect, in which there is an impact of the skin's own movements on the bone structure. These movements were not recorded by the system, leading to loss of information. This observation can be confirmed if we continue to compare values obtained by the three measurements in the remaining variables, where little change was observed among them, despite the different degree of extension.

Nonetheless, and despite the loss of information, we observed that the more the extension, the higher the values obtained in the remaining variables. This confirms the relationship between the WM and rotational movements in the lower extremity. This relationship has less impact as we move up the lower limb.

*Kappel-Bargas et al. (1998)* evaluated the relationship existing between extension of the 1st MTPJ and hindfoot movement during gait in the WM. Their results confirmed that in some subjects the MLA lifted immediately after joint extension, whilst in others, elevation was significantly delayed. This coincides with these subjects presenting more eversion of the calcaneus during gait (*Kappel-Bargas et al., 1998*).

For its part, unlike that suggested by *Kappel-Bargas et al. (1998)*, *Aquino & Payne (2001)* study, with a sample of 39 subjects with excessive hindfoot pronation, aimed to analyse the effect of excessive pronation on the WM during gait and list a series of static clinical measurements with the WM in dynamic state. This study reported that 15.4% of feet were classified as excessively pronated and that there was no statistically significant difference between excessive pronation and visual establishment of the WM. Despite this, *Aquino & Payne (2001)* considered, just like *Kappel-Bargas et al. (1998)*, that the location of the STJ axis, together with other measurements, has an impact on the implementation of an effective WM. This data was also referred to in the study by *Cintado et al. (2013)*, which evaluated the WM as an aspect that stabilised the forefoot and confirms our theory over the importance of a normalised STJ axis (*Kirby, 2001*) for it to function correctly. Hence this has formed part of the inclusion criteria in our research work.

*Cheng et al. (2008)* built a 3D model of finite aspects, where the foot was stimulated to ascertain the behaviour of the PF when stretched. The degree of contribution of the WM and strength of contraction of the Achilles tendon was evaluated. The results of this study suggested that increased tension in the PF is directly proportional to the increased degree of tension. The strength of the Achilles tendon also increased, which confirmed the results obtained by *Carlson, Fleming & Hutton (2000)* who evaluated how the extension angle of the 1st MTPJ affected the Achilles tendon—PF relationship. *Cheng et al. (2008)* also demonstrated that the maximum tension during PF stretching is concentrated around the medial tubercle of the calcaneus, and under the head of the first metatarsal (*Cheng et al., 2008*).

At the high tension recorded under the head of the first metatarsal by *Cheng et al. (2008)*, the results obtained by *Caravaggi et al. (2009)* that reproduced the WM by means of a 3D model using the combination of ultrasound and biometry during the gait support phase are added. A linear relationship was obtained between the maximum tension able to be tolerated by the PF and extension of the 1st MTPJ.

The results provided by *Cheng et al. (2008)* and *Caravaggi et al. (2009)* were very relevant because the high tension located under the head of the first metatarsal may be associated with compression of the soft tissues present in the area, mainly the gleno-sesamoid system, a possible causal factor among others of abnormality in our results in regard to extension of the 1st MTPJ.

The most important limitations found in this study refer to sample size and use of the inertial sensor system. Because they are not stuck to the skin, possible movements that occur between the bone segment and the skin are not quantified. This impact has an impact on the movements recorded and some of the information is lost.

As a new research line, we plan to increase sample size; in addition to trying to find a kinematic system able to quantify both degrees of extension of the 1st MTPJ, and rotational movements of the legs, where there is a minimal impact of soft tissues present in the area.

We could also establish a comparison among subjects that meet the normality criteria set and subjects with a tendency towards pronation, thereby enabling us to determine whether or not WM is efficient and its relationship with the foot's functionality.

## CONCLUSION

Kinematic analysis of the lower limb movements because of the WM activation suggest that the more the extension, the more movement will be generated, thereby reducing the level of impact the more distal the structure with respect to the joint. We also state that the soft parts, mainly the gleno-sesamoid system, have an impact on determination of articular movement. The fact of making an extension movement on the phalanx is not indicative of acting on the joint itself. An indirect measurement is therefore taken.

We can state that extension of the 1st MTPJ has an impact all over the legs, not only generating an external rotation movement of the tibia. However, because the kinematic system used was not suitable, its impact was not exactly quantified. A system that considers all factors that interfere with recording movement, such as the soft tissues, needs to be found or created.

### Funding
The authors received no funding for this work.

### Competing Interests
The authors declare that they have no competing interests.

### Author Contributions
- María José Manfredi-Márquez conceived and designed the experiments, performed the experiments, analysed the data, contributed reagents/materials/analysis tools, wrote the paper, prepared figures and/or tables, reviewed drafts of the paper.
- Natalia Tovaruela-Carrión conceived and designed the experiments, performed the experiments, wrote the paper, prepared figures and/or tables, reviewed drafts of the paper.
- Priscila Távara-Vidalón performed the experiments, contributed reagents/materials/analysis tools, reviewed drafts of the paper.
- Gabriel Domínguez-Maldonado performed the experiments, contributed reagents/materials/analysis tools, reviewed drafts of the paper.

- Lourdes María Fernández-Seguín performed the experiments, contributed reagents/materials/analysis tools, reviewed drafts of the paper.
- Javier Ramos-Ortega conceived and designed the experiments, performed the experiments, analysed the data, wrote the paper, prepared figures and/or tables, reviewed drafts of the paper.

## Human Ethics

The following information was supplied relating to ethical approvals (i.e. approving body and any reference numbers):

Comité Coordinador de Ética de la Investigación Biomédica de Andalucía (España), CEI de los Hospitales Universitarios Virgen Macarena y Virgen del Rocío, granted Ethical approval to carry out the study.

## Data Availability

The raw data has been uploaded as Supplemental Dataset Files.

## Supplemental Information

Supplemental information for this article can be found online at http://dx.doi.org/10.7717/peerj.4103#supplemental-information.

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
