# Peer review of "Three-dimensional variations in the lower limb caused by the windlass mechanism"

_PeerJ, doi:10.7717/peerj.4103_

## Round 0.1 · original submission · Major Revisions

We thank you for your submission of this manuscript at PeerJ. The reviewers have highlighted a number of issues with the initial submission of this manuscript that require careful attention. I would therefore suggest you carefully read these comments and consider how you can improve the manuscript to the standard required by the reviewers.

Of particular note, they have identified aspects of the written English as needing improvement. I would therefore recommend you get additional assistance on this aspect of the manuscript. We suggest you have a fluent, preferably native, English-language speaker thoroughly copyedit your manuscript for language usage, spelling, and grammar. If you do not know anyone who can do this, you may wish to consider employing a professional scientific editing service.

Thank you for considering PeerJ for publication of your manuscript.

·

Basic reporting

This article meet the standards of the journal

However, is necessary to check some expressions in English

Experimental design

1. It would be necessary to indicate the system of measurement of the anthropometric measurements (height, weight)

2. It would be helpful to indicate what of image was used to calculate the STJ with the Autocad system

3. Please, tell us something about the results of the test test of wilcoxon for the test with the independent sensors

4. What happened with the test in which the sensors were related ?

Validity of the findings

I think that the findings are accurate, valid and very helpful for the scientific community

Additional comments

This is a very good research and paper, I think it could be very cited in the podiatric medical community

·

Basic reporting

In general, edit for word tense and word choice. A native English spoken should review the whole paper for clarity before publication.

The manuscript is full of grammar and spelling mistakes that should be corrected.

Experimental design

There are major issues regarding design reporting in the manuscritp. Please, see the general comments for authors section.

Validity of the findings

Conclusions are well stated.

Additional comments

The paper is quite interesting but needs some modifications in order to be published. So, Some mayor issues and a few minor ones have to be addressed:


MAJOR

The major point of the paper is that the Materials and Methods are not clearly described in the paper which the manuscript not very understandable. This should be corrected before publication.

1. Kinematic description should be more clearly stated in the manuscript. The description of kinematics is usually quite complex and in this case needs a more detailed description in this paper. The starting point were the measurements were taken in the Bioval system should be clearly stated. You should define which was the zero position from the measurements taken. Did the system take a reference position from the relaxing standing position of the patient? Or did the system take the reference position from other choice?. For example: you say that metatarsophalangeal joint dorsiflexion was 8.83º with dependent sensors. Which was the reference position: standing position or 0º position?. The same can be applied to the all variables.
At the same time, from the tables, it seems that variables measured were extension of 1st MTP joint, plantarflexion of 1st metatarsal, prono-supination of first metatarsal, shank rotation, femur rotation and STJ. It should de define how this variables were measured with the Bioval system. Please, specify what STJ means and how was measured.
2. In the manuscript there is a lack of explanation of the variables measured and compared. The authors used a significance test but we do not know which variables were compared. In lines 152-153, authors stated “to determine whether statistical significance was attained we used the Wilcoxon test for two related samples and to evaluate the correlation between variables Spearman’s Rho test was used.” However, it is not known what comparisons did the authors. You used a Wilcoxon rank sum test but
you need to specify what exact variables did you compare and why and, also, this should match with the objectives of the study stated in the introduction. Lines 165 -168, please specify the values. Without this information, the results section is not understandable.
3. Why do you use the dependent sensors with the 45º wedge and the independent sensors with the maximum extension? From reading the manuscript it is not clear why you chose that methodology. If there is a reason for that, it should be clearly stated in the manuscript. Which is the difference between dependent and independent sensors?
4. Subtalar Joint Axis Location technique is highly subjective and you should state which investigator/investigators performed the test along with other information such us years of experience. Furthermore, How do you define normal? There is no clear agreement regarding what can be defined as a normal STJ axis using this test. I would recommend to state what would be considered abnormal and used it as a exclusion criteria.

MINOR:

In general, edit for word tense and word choice. A native English spoken should review the whole paper for clarity before publication.

ABSTRACT:
The methods section of the abstract should be more clearly define. It it not possible to have an understanding of the results of this interesting paper if you do not specify the method section better.

INTRODUCTION:
Regarding the introduction, authors give a huge explanation of the windlass mechanism. That description is so large and for easy reading I would recommend to shorten it. Furthermore, from the introduction I am not sure what gap(s) in the literature you hope this study can address. Please briefly tell the reader this before you declare a study purpose.

MATERIALS AND METHODS
Is it the statement in 106 right? (Oviedo Agreement and the Declaration of Helsinki? (Sure is Oviedo?)

In the exclusión criteria you stated, “inability to understand instructions related to the study”. As participants of the study were students from your university it seems hard to believe this exclusion criteria, please, modify it.

I recommend to change “variables measurement” instead of the “Methodology” subheading in the Materials and Methods section.

In order for reproducibility of the study you should say which investigator or investigators did the measurements. Was just one or more than one? These aspects should be clearly stated in the test so anyone can reply the study.

As Bioval system is not widely used or known, I would recommend a brief (1 or 2 sentences) description of the system.

I would recommend to add a subheading “Statistical Methods” in line 144.

Line 147, you stated “pronosupination of the first metatarsal”. I would recommend to clarify this concept as considerable confusion still exists regarding axis of movement and movement of first meatatarsal-ray in the literature.

It is recommended to say “Wilcoxon Rank sum test” instead of just “wilcoxon”

DISCUSSION

The discussion is interesting and well developed. There are several studies that that have also found less than 60º of 1st MTP dorsiflexion for normal walking in normal subjects. AS a suggestion I would recommend the authors to use these studies in the discussion.

Reviewer 4 ·

Basic reporting

I have a quite a few concerns with this study.
- The background work needs to be thorough and talk about what has already been done and what information is lacking with proper references.
- The author needs to focus on what's novel about this study and the significance of it.
- The methods described are not clear and need to be more detailed.
- Last but not the least, the language used throughout the paper is very hard to understand. The manuscript should be reviewed by an English language expert before any further consideration.

Experimental design

The experimental design is not clear and needs to be more detailed.

Validity of the findings

It is difficult to comment on the validity since the design of the study is not clear and the language used is ordinary.

Additional comments

I have a quite a few concerns with this study.
- The background work needs to be thorough and talk about what has already been done and what information is lacking with proper references.
- The author needs to focus on what's novel about this study and the significance of it.
- The methods described are not clear and need to be more detailed.
- Last but not the least, the language used throughout the paper is very hard to understand. The manuscript should be reviewed by an English language expert before any further consideration.

Thank you for submitting your work.

---

## Round 0.2 · accepted · Accept

We thank you for attending to the reviewers comments on your revised manuscript and am happy to let you know the paper has now been accepted for publication.

·

Basic reporting

Clear an unambiguous, professional

Experimental design

Original

Validity of the findings

OK